# Application of Copper Thermal Spraying for Electrical Joints between Superconducting Nb_3_Sn Cables

**DOI:** 10.3390/ma15010125

**Published:** 2021-12-24

**Authors:** Vincenzo D’Auria, Pierluigi Bruzzone, Mickael Sebastian Meyer, Enrique Rodriguez Castro, Stefano Sgobba

**Affiliations:** 1Ecole Polytechnique Fédérale de Lausanne (EPFL), Swiss Plasma Center (SPC), The Paul Scherrer Institute (PSI), CH-5232 Villigen, Switzerland; pierluigi.bruzzone@psi.ch; 2European Council for Nuclear Research (CERN), CH-1211 Geneva, Switzerland; mickael.sebastien.meyer@cern.ch (M.S.M.); enrique.rodriguez.castro@cern.ch (E.R.C.); stefano.sgobba@cern.ch (S.S.); 3Campus Leganes, University Carlos III of Madrid, Avenida Universidad 30, 28911 Madrid, Spain

**Keywords:** superconducting magnets, electrical joints, electric arc spraying, microstructure

## Abstract

This manuscript reports on the application of copper thermal spraying in the manufacturing process of an electrical connection between Nb_3_Sn cables for superconducting magnets of fusion reactors. The joint is realized through diffusion bonding of the sprayed coating of the two cables. The main requirement for such a connection is its electrical resistance, which must be below 1 nΩ at B = 8 T, I = 63.3 kA and T = 4.5 K. Micrographs of the joint prototype were taken to relate the joint resistance with its microstructure and to provide feedback on the manufacturing process. Optical microscopy (OM) was used to evaluate the grain size of the coating, presence of oxide phases and to analyze the jointed surfaces. Scanning electron microscopy (SEM) and, in particular, energy-dispersive X-ray spectroscopy (EDX) were used to confirm the elemental composition of specimens extracted from the prototype. It is shown that the copper coating has an oxide concentration of 40%. Despite this, the resistance of the prototype is 0.48 nΩ in operating conditions, as the oxides are in globular form. The contact ratio between the jointed surfaces is about 95%. In addition, residual resistivity ratio (RRR) measurements were carried out to quantify the electrical quality of the Cu coating.

## 1. Introduction

This work is within the framework of the European DEMO project (DEMOnstration power plant) [1], which aims to prove the reliable production of electricity from the fusion reaction of deuterium and tritium. This machine, the construction of which should begin around 2040 and its operation in 2050, is acknowledged as an important step in the Roadmap to Fusion Electricity [2,3], i.e. the document illustrating the strategy to get to this goal.

The DEMO fusion reactor is based on the Tokamak concept [4], a toroidal machine with several magnet systems that generate the magnetic field confining the hot plasma, constituted by deuterium, tritium and other elements. Among the magnet systems, in this manuscript we concentrate on the Toroidal Field coils (TF), i.e., the ones generating the toroidal component of the magnetic field. Currently, three research institutes in Europe are designing different TF options [5]. In this paper, we consider the TF design proposed by the Swiss Plasma Center (SPC) [6]. The intermetallic compound Nb_3_Sn [7] is the superconducting material considered for this magnet. This material belongs to the class of low temperature superconductors (LTS), i.e. materials that have to be cooled below 77 K (the boiling point of nitrogen at atmospheric pressure) or to lower temperatures to be superconducting. It is chosen for the TF design of the SPC on the basis of both the magnet peak field and its operating temperature, which are about 12 T and 4.5 K, respectively. Under these operating conditions, the cheaper NbTi is not an option because it is not superconducting. Moreover, magnets based on Nb_3_Sn have a technological maturity that is much higher than those based on high temperature superconductors (HTS), i.e., materials which are superconducting above 77 K.

In applications, the Nb_3_Sn precursors are present as filaments inside a copper matrix. The filaments and matrix form a wire. The surface of the wire is made of copper with high purity to stabilize the superconductor against thermal and electromagnetic disturbances. Pure copper is chosen for its particularly low electrical resistivity at cryogenic temperatures. Several of these wires are twisted together to form the cable (Figure 1).

The strands are plated with chromium in order to avoid sintering among them during the heat treatment of the cables. This cable is heat treated so that the precursors form the superconducting Nb_3_Sn through a solid-state diffusion process [8]. After that, the cable becomes brittle and is able to carry 63.3 kA without losses at 4.5 K and 12.23 T. Such a cable is at the base of the winding of the TF magnet proposed by the SPC. Further details about this and the other components of the conductor can be found in [9,10].

The adjacent layers of the TF winding are electrically connected in series through electrical joints, which are the resistive components of the superconducting magnet and topic of this manuscript. The joint prototype development is reported in [11]. It is based on the diffusion-bonding [12,13] of overlapped cables. Therefore, it is based on good contact between the two cables. For this reason, the wavy surface of the cable is coated with copper and milled flat. Copper is chosen for its low electrical resistivity at cryogenic temperature. The selected copper coating technique is electric arc spraying [14], as it is faster than other coating techniques, e.g., electroplating, and the cheapest and most simple among thermal spraying techniques. A temperature range of 600–700 °C and a pressure of some tens of MPa [15] are required to bond the Cu surfaces in contact. The temperature in particular is comparable to the one that the cable has already experienced during the reaction heat treatment to form Nb_3_Sn. The steps for the joint fabrication are summarized as follows [11]:The surfaces of the cables are sandblasted to roughen the surfaces in view of the thermal spraying and the removal of the chromium layer, which is originally deposited on the wire to avoid the sintering of the strands during reaction heat treatment and to reduce AC losses in operation [16];The cables are coated with copper;The surfaces are machined flat;A clamp applies pressure on the surfaces of the joint;The region is locally heated with an inductive oven, while nitrogen gas provides an inert atmosphere.

A cross-section of the joint prototype is reported in Figure 2. The successful electrical test motivated to carry out metallographic analyses aimed at a further characterization of the joint in order to link its microstructure to the electrical performance is reported in a previous publication [11]. The regions of interest are the wire-coating and coating-coating interfaces, as well as the bulk of the coated layer. The types of analyses and their results are presented in this manuscript.

## 2. Methodology: Copper Thermal Spraying Applied to the Nb_3_Sn Cables

Before thermal spraying, the cable is held with an aluminum fixture that limits the cable deformation induced by the thermal gradient during the coating process. Such a deformation was experienced in previous sample preparations, as shown in Figure 3 for dummy cables made of copper.

As this fixture partially covers the surfaces to be coated, sandblasting and arc spraying are performed in multiple steps:Sandblasting is applied;Copper is sprayed on the free surfaces of the cable until 1 mm is deposited;The Al supports constraining the cable are removed. The deposited hard copper layer can act as constraining support, limiting the cable deformation;The remaining surface sections are sandblasted;The overall length of the joint is sprayed again until a maximum thickness of 3 mm is reached.

In Figure 4, the coating procedure is summarized.

The copper wire used for the arc spraying process has a diameter of 1.58 mm. Its purity was measured in our facility in terms of residual resistivity ratio (RRR). This turned out to be 43 for the raw wire and 200 after its annealing. For the Cu molten particles, a value between these two is expected. The temperature of the electric arc was about 4000 °C [17], whereas the surface temperature of the cable, monitored during the spraying process, was below 60 °C. The electric arc spraying procedure was performed in an environmental atmosphere. Further information can be found on the website of the supplier [17]. The coating is milled flat until 1 mm remains on the joint side. After spraying, the surface of the coating is rough, opaque and the shape preserves the waviness given by the strands. When milling, the surface becomes flat, shiny and smooth. The cables are then inserted into the clamp that applies pressure while the inductive oven heats the region locally in an inert atmosphere, as described in [11]. The generator that feeds the inductive coil worked at a power of 4.6 kW to reach the required steady-state temperature and keep it. The AC current frequency in the coil was 52.3 kHz. The temperature of the cable was 690 °C at the center of the clamp and 620 °C at the side of the clamp. The reason behind the temperature gradient is the cable cooling at 20 °C with a 250 mm distance between heated and cooled regions. An N_2_ dewar provides the gaseous inert atmosphere. A picture of the diffusion bonding set-up is shown in Figure 5. Further information can be retrieved from [11].

Three types of samples were taken for the micrographic studies (Figure 6):A cross-section of the cable before the thermal spraying process (Sample #1);A cross-section of the cable after the copper spraying (Sample #2);Two cross-sections from the electrically tested joint (Samples #3,4).

All cuts were performed by electro-erosion. The rational of this series of samples is to track the evolution of the micrographic structure during the joint manufacture. The approach includes optical microscopy, scanning electron microscopy and energy-dispersive X-ray spectroscopy.

## 3. Results

The digital microscope (VHX-6000 from the Japanese Keyence) images highlighted the characteristics of the strands for all samples, the strand–coating interface for Samples #2–4 and the coating–coating interface for Samples #3,4. The results are the following:Some residual chromium was found at the strand–coating interface in Sample #2 (Figure 7), thus meaning that the sandblasting procedure did not remove it completely.A porosity concentration in the range 5–10% was found in the Cu coating of Samples #2–4. They appear as black spots in Figure 7, Figure 8 and Figure 9.The EDX analysis revealed the presence of copper oxides in the Cu coating, present as a grey color in the images taken at the digital microscope. They were in elongated form in Sample #2 and, after diffusion-bonding, were in globular form (Samples #3,4), as illustrated in Figure 8. An oxide concentration of 40 ± 5% was determined for Samples #3,4 through image analyses based on color thresholding.At the coating–coating interface, i.e., at the diffusion-bonded surface of Samples #3,4 the contact ratio is 94.5% for Sample #3 and 96.3% for Sample #4. To determine these numbers, a series of magnifications along the contact line of the two cross-sections was taken (Figure 9). In these, the length of voids and pores along the contact line was subtracted from the line length.The contact between strand and coating is not homogeneous (Figure 9c). In particular, the strands in the interstices between adjacent sub-cables show less contact with the Cu coating.

The grain size of the coating after diffusion bonding was analyzed (Figure 10). For this purpose, one of the diffusion-bonded samples was etched with the etchant #34, as per ASTM 407, to remove the impurities. The optical microscope Axio-Vision from the German Zeiss was subsequently used to evaluate the grain size of copper according to ASTM 112, which resulted in G equals to 13.5 (3.3 μm average diameter of the grain size, as shown in Figure 9). The micrographic analyses were complemented by RRR measurements of the copper coating carried out at SPC-Villigen (Switzerland). Three samples were extracted from a dummy joint (Figure 11) i.e., built with copper cables using the same procedure as for the joint between Nb_3_Sn cables. The resistance of the samples was measured along their length at room temperature and 4.5 K. The RRR, i.e., the ratio between the two measured resistances, was 70 ± 5. The same type of measurement was carried out for the wire melted to deposit the coating. Before and after annealing in an inert atmosphere, the RRR of the wire was 43 and 200, respectively.

## 4. Discussion

It was mentioned that Cr residues are present on the surface of the strands to be coated with copper, despite the sandblasting procedure (Figure 7). Nevertheless, they are not expected to influence the joint resistance, as their dimension is in the μm range. They should not even affect the adherence between strand and sprayed copper because in the thermal spraying process the bonding is not chemical but physical, as the melted particles get stuck in the interstices of the surface roughened through sandblasting.

It is shown that the contact between strands and copper coating is not homogeneous (Figure 9c). This occurs because the interstices between sub-cables are the deepest points and are therefore more difficult for the molten particles to access during the thermal spraying procedure.

The porosity concentration in the coating is in line with the values given by the thermal spraying company, Bührer AG [18]. The oxides concentration in the coating is instead 20–30% higher than the values declared by the producers (Figure 8). We tend to exclude that such a difference arises during the diffusion bonding process, in which an inert N_2_ atmosphere is provided. In addition to this, the samples taken before and after diffusion bonding show a comparable oxide concentration, as it can be noticed in Figure 8. Moreover, no inert atmosphere was present during the spraying process. Such a high oxide concentration limits the growth of the copper grains, as shown in Figure 10, despite the coating experiencing more than 600 °C for more than two hours during the diffusion bonding treatment.

Despite the high oxide concentration and the limited copper grain size, the RRR value of the coating is elevated and the electrical resistance of the joint is low and within the allowable limit. Such a result is possible because the oxides transform from the elongated form of Sample #2 into the globular one of Samples #3,4 which favor the copper connectivity.

The coating–coating interface, i.e., where the two cables are diffusion-bonded, presents non-contact regions that limit the diffused surface (Figure 9). The extension of the non-contact regions highly depends on where the cut of the cross-section is performed. In fact, our interpretation is that the contact ratio between the two coated flattened surfaces strongly depends on the machinability of the coating, which we observed left grooves on the surface while approaching the strand during machining (Figure 4c). For this reason, we did not go below 1 mm thickness during the milling of the coating, as we experienced high degradation of the surface flatness.

## 5. Conclusions

The application of the arc spraying technique for the manufacture of an electrical connection between superconducting cables was presented in this manuscript. Its excellent resistance performance, documented in a previous publications, was confirmed in this paper with regard to the microscopic structure of the joint and, in particular, the copper coating. It was demonstrated that the employment of the arc spraying technique in a non-inert atmosphere is able to provide the required low electrical resistance, despite oxidation. However, even lower electrical resistances can be obtained if this is necessary in other applications. This could be achieved either by providing an inert atmosphere during the copper arc spraying process or by the employment of a thermal spraying technique characterized by lower oxide concentration, e.g., plasma spraying or high-velocity oxygen fuel (HVOF). The coating realized by these other two spraying techniques might also enhance the machinability of the surface, thus improving the contact ratio in the joint. In fact, the coating of both plasma spraying and HVOF are in general characterized by a higher pull strength [18]. As the joint has to withstand the electromagnetic forces arising during the magnetic operation, a test will be designed in the future to characterize its mechanical strength.

## Figures and Tables

**Figure 1 materials-15-00125-f001:**
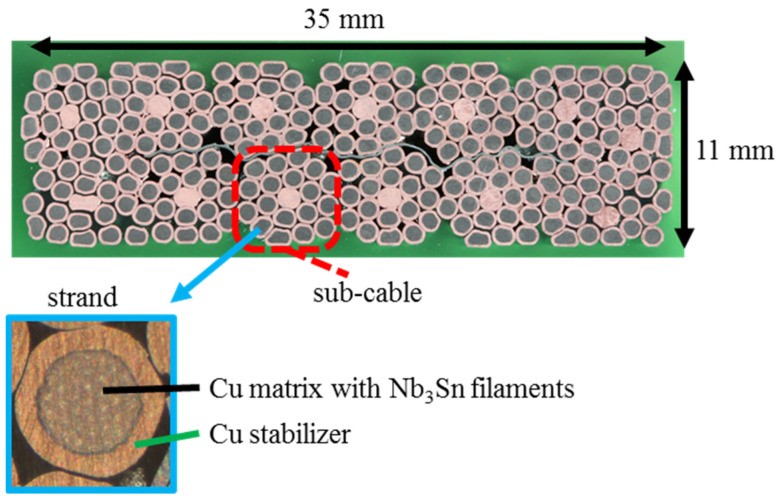
Cross-section of the superconducting Nb_3_Sn cable used in the joint manufacture.

**Figure 2 materials-15-00125-f002:**
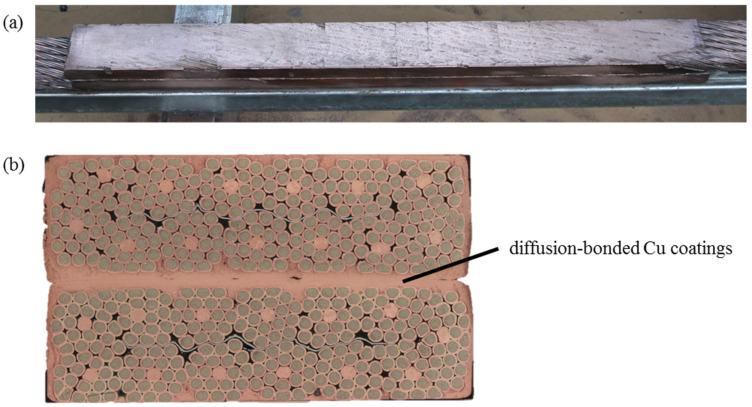
Diffusion-bonded joint between Nb_3_Sn cables coated with copper thermal spraying (**a**) and cross-section of the joint (**b**).

**Figure 3 materials-15-00125-f003:**
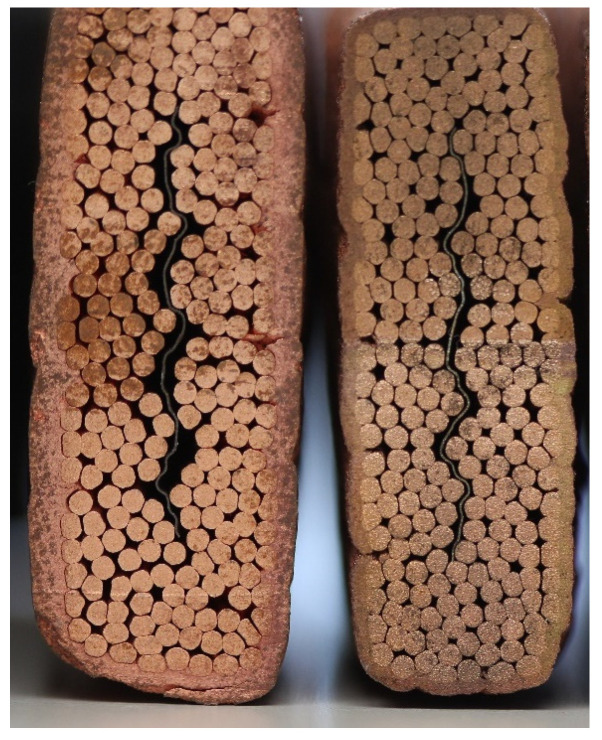
Comparison between two dummy cables after Cu thermal spraying. On the left, the cable was coated as free-standing. On the right, an Al fixture was constraining it during coating.

**Figure 4 materials-15-00125-f004:**
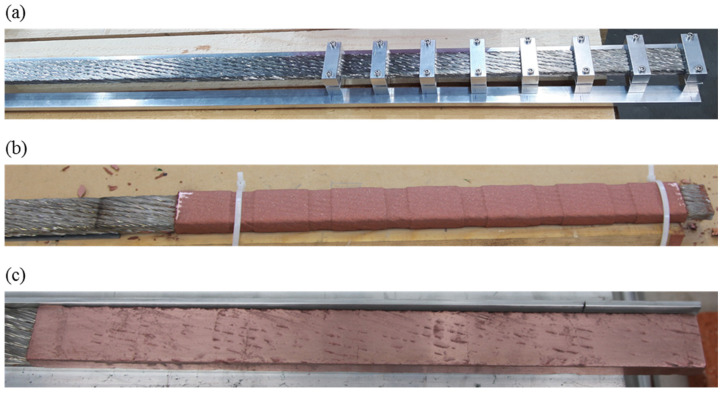
Superconducting cable before Cu arc spraying (**a**), after Cu deposition (**b**) and milling of the surfaces (**c**).

**Figure 5 materials-15-00125-f005:**
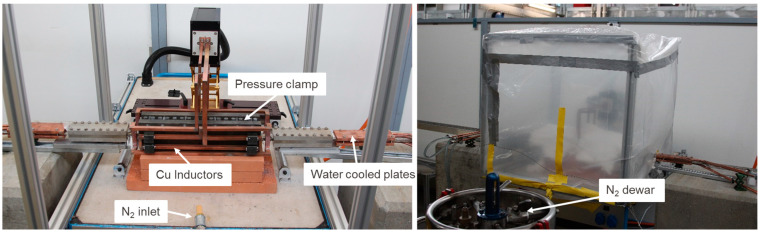
Diffusion bonding set-up during assembly (**left**) and during operation (**right**).

**Figure 6 materials-15-00125-f006:**
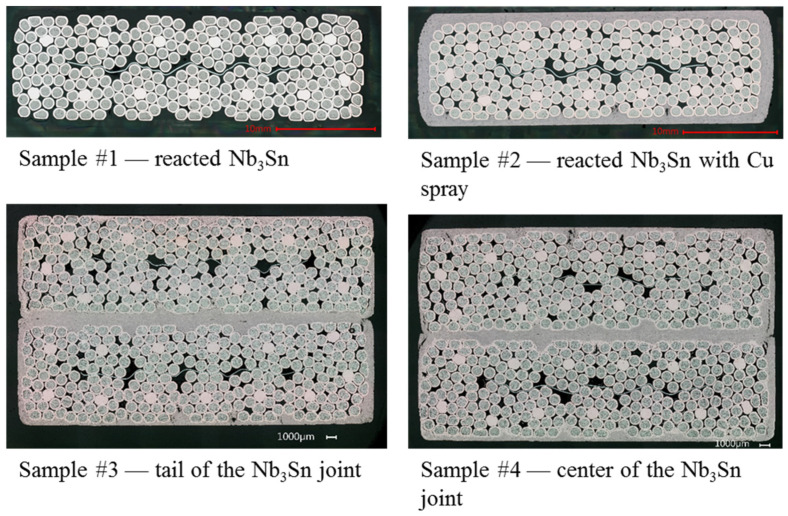
Samples for the micrographic analysis.

**Figure 7 materials-15-00125-f007:**
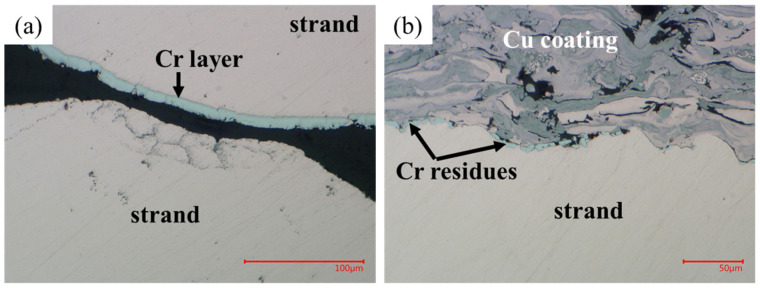
Comparison between the Cr layer of an internal strand (**a**) and the Cr residues on a strand on the surface of the cable after sandblasting and Cu spraying (**b**).

**Figure 8 materials-15-00125-f008:**
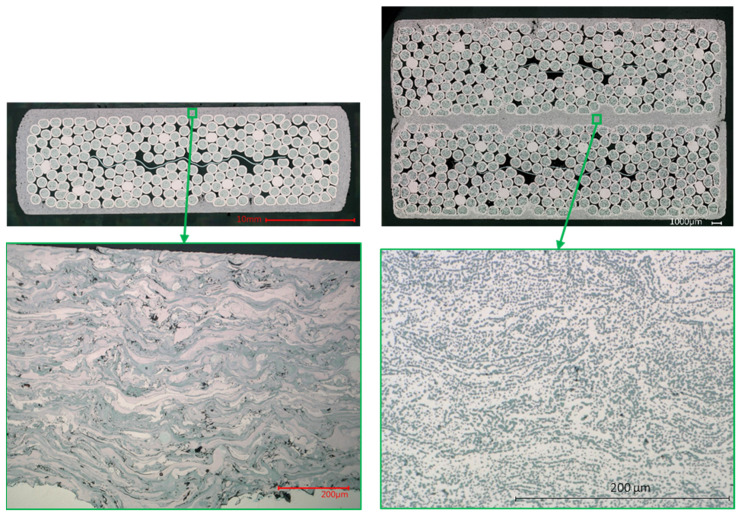
Form of the Cu oxides before (**left**) and after (**right**) the diffusion-bonding heat treatment.

**Figure 9 materials-15-00125-f009:**
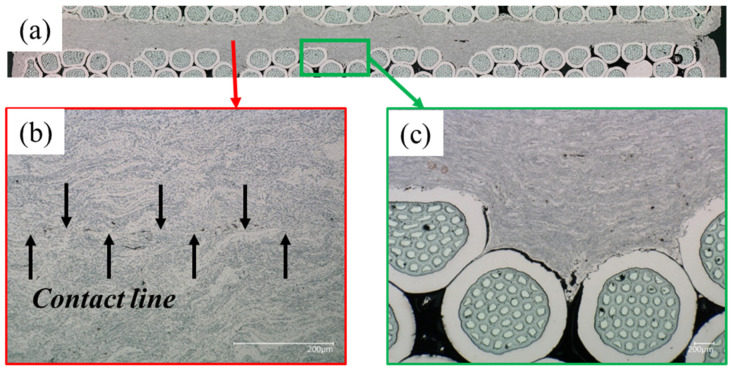
Diffusion-bonded joint of the copper coating of the two Nb_3_Sn cables (**a**), magnification of the contact line between the coatings (**b**) and detail of the detachment between one strand and the copper coating (**c**).

**Figure 10 materials-15-00125-f010:**
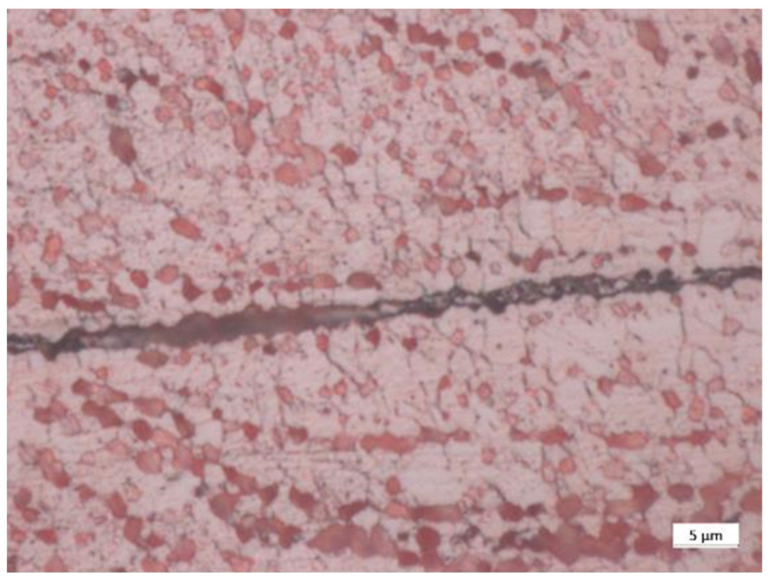
Detail of the grain size of copper next to the contact line of the diffusion-bonded surfaces after the chemical etching.

**Figure 11 materials-15-00125-f011:**
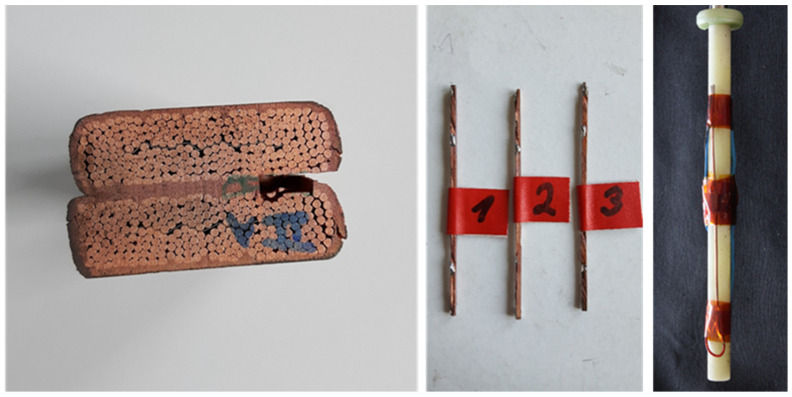
Samples extracted from the Cu coating of a dummy joint for the RRR measurements.

## Data Availability

The data presented in this study are available on request from the corresponding author. The data are not publicly available due to the large size.

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
