# Peer review of "Application of Copper Thermal Spraying for Electrical Joints between Superconducting Nb3Sn Cables"

_materials, 2021, doi:10.3390/ma15010125_

Round 1

Reviewer 1 Report

The paper is interesting and well written, suitable for publication in the submitted journal. 

Author Response

Thank you very much for your comments.

Reviewer 2 Report

Dear Authors:

In my viewpoint, the manuscript number1495495 v1, titled “Application of Copper Thermal Spray for Electrical Joints Superconductor Magnets for Fusion Energy” can be accepted to publication after minor correction. This ranking is justified below.

As a whole, I will comment each major topic of manuscript. In this sequence, I would like to highlight the strong technological aspect of manuscript instead academic one. In this sense, at moment, I suggest further valorization of aspects academics along of manuscript.

Title:

This title is improper, since contain aspects that weren’t approached in the manuscript; as an example “Fusion Energy”. Then, Fusion energy term should be deleted from Title. In same case, it is possible to mention Superconducting Magnets. Then, Superconducting Magnetic also should be deleted. A concise title would be as follow:

Joining functional Nb3Sn Cables via Copper Deposition based on the Thermal Spray Technique

The term functional is ascribed to the main requirement of B, I and T.

Abstract:

I suggest that Abstract be enhanced to accommodate some academic aspects correlated to very high magnetic field, electric current and very low temperature of operation. Also, I felt a bit the absence of standard. The first phrase and the last one sound strange. Again, the first phrase has a word that doesn’t belong to investigation that is superconducting reactor, then I suggest

The application of copper thermal spray in the manufacture process of an electrical connection between Nb3Sn superconductor cables is investigated.

The last phrase exhibits some kind of erroneous information. In the item conclusion should be exist only conclusion, any discussion. I suggest strongly change of phrase. In fact, this phrase would to finish as… resistance improvement is discussed.

Introduction:

In my opinion, the item Introduction needs be re write. See, one of function of this item is provide the state-of-art of a specific problem and highlight new contribution in the area of problem. At moment, manuscript’ text of the introduction is a truncated text or similar to this.

Also, the insertion of figures in the Introduction item is uncommon. I suggest that Figure 1 be moved at methodology. Otherwise, I suggest constructing the idea in sequence of capability of a single Nb3Sn, a sub-cable and the belt 35 mmm x 11 mm.

Methodology

As a whole, from Figure 1 there is an absent information at about the core of “sub-cable” that appears in a clear way in Figure 5.

In the Item results, I suggest that change the word pore by the void.  Therefore, voids between sub-cable are quasi an intrinsic feature, in the sequence voids size are increased after Cu spray should be discussed. As a matter of fact, as a function of coalescence of voids a great crack is formed at center of the belt. Here, Cu spray seems to undergo a kind of shrinkage.

Some careful with formal language is necessary. In this sense, one Figure show, any Figure summarizes.

Discussion:

Figure 4 should be revisited. The major aspect is that from Fig 1 any part of “belt” is a shiny metal. Fig 1 suggests that only “copper wish” surface would be observed. I suggest that from of the start of the text until the middle of text undergone further revision.

Results:

In this item, the presence of chromium is unclear. But, in the Fig 6 (a) is clear that “strand” can be recovered in a perfect way with Cr. The voids can be stemming another factor mismatch of thermal conductivity, wettability, vapor and gases deposited spontaneously, etc… I suggest that discussion be expanded. But, the shrinkage of Cu deposited by thermal spray seems a fact.

Conclusion:

In a broad sense, the conclusion is an item dedicated to the enhancement of the state-of-art approached in the item Introduction. Therefore, any discussion and reference should be replaced in the item Results and Discussion.

Reviewer 3 Report

This article focuses on applying copper thermal spray in the manufacturing process of an electrical connection between Nb3Sn cables for superconducting magnets in fusion reactors. The work is relevant to understanding and improving toroidal field coil (TF) technology. The Nb3Sn intermetallic compound chosen as the semiconductor material fits perfectly with current materials studied globally and is an excellent choice as filaments within a copper matrix.
Overall the work is well written, making it a pleasant and well-structured read. As a reader, I felt guided by the authors throughout the text. However, I have two suggestions for the authors to improve the manuscript. First of all, I think it is of paramount importance to add the intermetallic compound Nb3Sn to the work's title. Second, I ask for a greater focus on explaining the importance of the materials used (Cu and Nb3Sn) compared to other materials used in the literature.
I am delighted with the manuscript presented, and after these minor adjustments, the article is recommended for publication.

Author Response

We thank the reviewer for her/his comments. We implemented the proposed modifications in the introduction in the range of lines 43-55 and in line 68, in which we discuss about the choice of Nb3Sn as superconductor and copper as sprayed material.

Reviewer 4 Report

The authors present a thorough investigation of the material properties of copper coatings, which are required for the manufacture of superconducting toroidal coils for the magnetic field for plasma inclusion in a Tokamak setup.

They apply insight-driven basic experiments of metallographic characterization to a superconducting wire ensemble of practical relevance. They do not only analyse the features of the produced coatings but also suggest alterations of the procedure in order to improve the quality of the films and, thus, the manufactured electrical joints.

In conclusion, the manuscript describes a solid piece of work on material composite system with practical relevance and should be published in materials.   

Author Response

Thank you very much for your comments.